# Phenology, Seasonal Abundance, and Host-Plant Association of Spittlebugs (Hemiptera: Aphrophoridae) in Vineyards of Northwestern Italy

**DOI:** 10.3390/insects12111012

**Published:** 2021-11-09

**Authors:** Nicola Bodino, Stefano Demichelis, Anna Simonetto, Stefania Volani, Matteo Alessandro Saladini, Gianni Gilioli, Domenico Bosco

**Affiliations:** 1CNR–Istituto per la Protezione Sostenibile delle Piante, Strada delle Cacce, 73, 10135 Torino, Italy; nicola.bodino@ipsp.cnr.it; 2Dipartimento di Scienze Agrarie, Forestali e Alimentari, Università degli Studi di Torino, Largo Paolo Braccini, 2, 10095 Grugliasco, Italy; stefano.demichelis@unito.it (S.D.); matteo.saladini@unito.it (M.A.S.); 3Agrofood Lab, Dipartimento di Ingegneria Civile, Architettura, Territorio, Ambiente e di Matematica, Università degli Studi di Brescia, 25123 Brescia, Italy; anna.simonetto@unibs.it (A.S.); gianni.gilioli@unibs.it (G.G.); 4Agrofood Lab, Dipartimento di Medicina Molecolare e Traslazionale, Università degli Studi di Brescia, 25123 Brescia, Italy; stefania.volani@gmail.com

**Keywords:** color morphs, grapevine, host plant selection, insect vectors, *Philaenus spumarius*, plant preference, *Xylella fastidiosa*, xylem-sap feeders

## Abstract

**Simple Summary:**

Spittlebugs are the most abundant and widespread xylem-sap feeder insects in Europe. They are also the only proven vectors of the notorious bacterium *Xylella* *fastidiosa* (*Xf*) in the Old World. *Xf* inhabits the xylem of hundreds of plant species and is the causal agent of severe diseases to several crop plants, including grapevines. Since the spread of *Xf* depends on insect transmission, the study of vector abundance and ecology in the vineyard is of key importance in assessing the risk of disease spread to grapevines. The aim of this work was to gain information on *Xf* vector ecology and abundance in the vineyard agroecosystem. Herbaceous cover of inter-rows and headlands was colonized in spring by nymphs and in late summer/autumn by adults for oviposition, while woody hosts (grapevines and trees in the surroundings) represented a refuge during summer. Three spittlebug species were collected: *Philaenus spumarius*, *Neophilaenus campestris*, and *Aphrophora alni*, but the latter two species were very rare on the grapevine canopy. The presence of *P. spumarius* for an extended period on the grapevine canopy, together with its known ability to transmit *Xf* to grapevine, underlines the importance of preventing the introduction of *Xf* in *Xf*-free areas and of monitoring health conditions of grapevines in the *Xf*-infected areas of Europe.

**Abstract:**

Spittlebugs (Hemiptera: Aphrophoridae) are the vectors of the bacterium *Xylella fastidiosa* (*Xf*) in Europe. *Xf* may cause severe epidemics in cultivated plants, including grapevines. To assess the threat represented by the bacterium to grapevines, detailed information on the vectors’ phenology, density, and ecology in vineyards is needed. The aim of the present work was to describe spittlebug diversity, phenology, and host-plant association in the vineyard agroecosystem. Two separate field surveys of nymphal and adult spittlebug populations, i.e., a two-year survey of a single site and a one-year survey of three sites, were performed in vineyards of northwestern Italy in three consecutive years. *Philaenus spumarius* was the most common species, reaching average nymph densities on herbaceous cover up to 60–130 nymphs/m^2^. Adults were sampled on grapevines from May to September, with a peak in June (up to 0.43 insects/sweep). Herbaceous cover was colonized after egg hatching and in late summer for oviposition, while wild woody hosts represented a refuge during summer. The results show that spittlebugs can reach high population levels in vineyards, at least in the areas where the ground is covered by herbaceous plants for the whole season and the use of insecticides is moderate. The extended presence of *P. spumarius* adults on grapevines represents a serious risk factor for the spread of *Xf*. The scenarios of *Xf* establishment in vineyards in northwestern Italy and Europe are discussed in relation to the abundance, phenology, and plant association of spittlebugs.

## 1. Introduction

The anthropic introduction of exotic organisms into new environments is a phenomenon that has increased exponentially worldwide in recent decades, and, from a plant pathology perspective, is leading to a significant increase of new outbreaks of agricultural and forest pathogens, which are often insect-transmitted [1,2]. In certain cases, pathogen introduction may lead to the establishment of novel associations with native vector species, possibly changing the status of former minor pest species to major pests in some agroecosystems [3,4].

The introduction of the xylem-limited bacterium *Xylella fastidiosa* Wells (*Xf* hereafter) in recent years to Apulia (southern Italy), which resulted in a dramatic dieback of olive trees, focused attention on a previously overlooked native insect taxon in agricultural research in Europe: the true spittlebugs (Hemiptera: Aphrophoridae). Following the discovery of *Xf* in southern Italy, intensive research efforts have been carried out to investigate different aspects of the biology and ecology of the *Xf* insect vectors in Europe, mainly in the olive agroecosystem. Spittlebugs, and in particular the meadow spittlebug *Philaenus spumarius* (L.), have been recognized as the insect vectors responsible for the spread of the bacterium among olives and other host plants in different Mediterranean ecosystems in Italy, Spain, and France [5,6,7,8].

*Xf* is characterized by a great genomic diversity among subspecies and strains, resulting in a very wide host-plant range, and it is associated with several plant diseases, including Olive quick decline syndrome (OQDS), Citrus variegated chlorosis (CVC), Almond leaf scorch (ALSD), and Pierce’s disease (PD) of grapevine [9]. Moreover, the possibility that the *Xf* genotype associated with OQDS may infect grapevine was not excluded by an ad hoc risk assessment [10]. Pierce’s disease—associated with *Xf* subsp. *fastidiosa* ST1—is a major threat to grapevine production in California [11,12]. Since the identification of *Xf* as the causal agent of PD, research on vectors has focused mainly on sharpshooters (Hemiptera: Cicadellidae: Cicadellinae), the most common xylem-sap feeders in the Americas [13,14,15]. Conversely, the spittlebugs have long been considered of little importance, given their low abundance compared to sharpshooters in the investigated agroecosystems [16,17,18]. Recent studies have highlighted the possible role of spittlebugs in *Xf* transmission on grapevines and consequently on PD incidence and epidemics in vineyards, at least in some wine-producing areas of California [19,20,21].

*Xf* has been considered a major threat to the European wine sector for decades, being a quarantine organism since 2000 [22]. The surveys in EU countries prompted by the discovery of the pathogen in southern Italy revealed different past introductions of *Xf* in the Mediterranean region [23,24]. The *Xf* strain associated with PD (ST1) was found on grapevines in the island of Mallorca (Spain) in 2016, although it has probably been there since the last decade of the 20th century [24,25]. The presence of this pathogen in a very restricted vine-growing area has gone unnoticed probably due to its quite low incidence, with it not causing relevant economic losses [26]. However, the presence of *Xf* ST1 in Mallorca is of great concern for European viticulture, as is the risk of its further introductions from the Americas to other European vine-growing areas.

To our knowledge, information available on spittlebug ecology in European vineyards is currently scarce. Spittlebugs are common and often abundant in organic vineyards [27,28,29,30,31]. *Philaenus spumarius* is the main spittlebug species found on grapevines—especially in mid-June—and on both herbaceous and woody plants inside and in the areas surroundings vineyards [31,32,33,34]. However, no targeted studies on xylem-sap feeder population dynamics in the vineyard agroecosystem have been conducted, even though information on the presence, abundance, and ecology of spittlebugs is crucial to quantitatively estimate the risk of *Xf* outbreaks in wine-producing areas and to design potential effective control measures against the vectors. In this study, we present the results of two field surveys: (i) a two-year survey (2016–2017) of spittlebug populations in an organic vineyard in northwestern Italy, with a focus on the abundance, phenology, and host-plant preferences of both nymphal and adult stages of these xylem-sap feeders, (ii) a survey performed in 2018 in the same vineyard monitored in the previous survey and in two other organic vineyards in the same region.

## 2. Materials and Methods

### 2.1. Survey Sites

The two-year survey was carried out in a 12-year-old multivarietal organic vineyard in the province of Asti (Piedmont Region, Italy) in 2016–2017 (Azienda “Campo sperimentale Viatosto”—CREA) (N 44.921740, E 8.195633 (ca. 2.3 ha)) (Figure 1). The grapevine cultivars were Albarossa, Barbera, Chardonnay, Cortese, Incrocio Manzoni, and Syrah. The vineyard was southeast facing, and the rows were oriented northeast to southwest. The surroundings were composed of an abandoned vineyard to the west, woods with rootstocks to the east, and strips of woods (width ca. 10–50 m) at north and south-southeast, separating the vineyard from surrounding meadows. Woods were composed of mixed broadleaf species (mainly *Quercus petraea* L.). The vineyard ground cover was composed of spontaneous gramineous grasses and broadleaf species. The surveyed vineyard was selected based on low-input management, e.g., no synthetic insecticides were sprayed, and any tillage or mowing was performed during spring (the nymphal development period) in 2016, while in 2017 a single mowing was carried out in April. Spring weeding on the row and three insecticide treatments with pyrethrum (applied in June–July) against *Scaphoideus titanus* Ball were carried out in the vineyard during the field survey, possibly interfering with vineyard colonization by insects.

During 2018, a second survey was carried out in the Asti vineyard and in two other organic vineyards in Piedmont (Cisterna (CdA): N 44.825894, E 8.009331 (ca. 1.1 ha); Paderna (PD): N 44.826167 E 8.895371 (ca. 0.701)) (Figure 1), to provide a more comprehensive picture of the spittlebug population levels in the organic vineyards of the region.

### 2.2. Sampling of Nymphal Stages

Nymphal populations of spittlebugs and their host plants were monitored in the vineyards and the immediate surroundings. Nymphal stages were counted using the quadrat sampling method, which is frequently used to quantify the abundance of spittlebug juveniles [6,8,35,36]. Quadrats (secondary sampling unit for preimaginal instars: SSU_p_; 0.25 m^2^—100 × 25 cm each) were randomly positioned on the ground cover within the primary sampling unit (PSU) at each sampling date. Since nymphal sampling is partially destructive, the SSU_p_ locations were always different on different sampling dates.

Vegetation and soil surface inside the quadrat were visually inspected for at least five minutes for the presence of spittlebug nymphs. The latter were identified at the species level and their preimaginal stage determined. Spittlebug samplings were conservative and instar was determined in the field using classification from Vilbaste [37]. A few nymphs from each vineyard were collected and reared to adult stage in laboratory facilities to confirm species identification.

Ground plant community was defined for each SSU_p_ based on (i) average height of herbaceous plant cover, (ii) total plant cover percentage, and (iii) cover percentage of the most abundant plant taxa (two to four dominant plant taxa/SSU_p_). Plants found with spittlebug foams (i.e., host plants) and the most abundant plants inside the SSU_p_ were identified at the genus level in the field or sampled for later identification in the laboratory using classification keys from Pignatti [38]. Since field samplings started in late winter, plants were often not yet showing the characteristic morphological traits necessary for a correct identification at the species level. The phenological phases of host plants at the time of sampling were classified into three classes: preflowering, flowering, and postflowering phases.

In the 2016–2017 survey, the primary sampling unit (PSU) was the Asti vineyard and surroundings (ca. 3.8 ha). A total of 30 quadrats were sampled within the inter-rows, while 30 quadrats were sampled on the ground cover of headlands surrounding the vineyard, for a total of 60 SSU_p_ locations in the whole PSU at each sampling date. The sampling of nymphal stages was carried out weekly or fortnightly from early March until late May or until no nymphs were found in any SSU_p_ for two consecutive sampling dates. In the 2018 survey, 15 quadrats (SSU_p_) were sampled within each vineyard on a single date in mid/late April.

### 2.3. Sampling of Adult Stage

Spittlebug adults were sampled using a sweep net (38 cm in diameter) on three different vegetation types inside the PSU: herbaceous cover, grapevine canopy, and wild woody plants (shrubs or trees growing in the immediate surroundings of the vineyard). Sweep netting has proven to be the most reliable sampling method for the adult stage of spittlebugs [6,8,39]. Samplings were carried out at random locations within PSU for each sampling date, to avoid repeated disturbance on the same points during the sampling season.

In the ground vegetation, spittlebugs were sampled in randomly distributed secondary sampling units (SSU_h_), consisting of four sweeps each performed along a 2.8 m transect. Therefore, the area effectively sampled by sweeping in an SSU_h_ was estimated as about 1.0 m^2^ (0.7 m length × 0.38 m width × 4 sweeps). On the grapevine canopy, insects were sampled on randomly distributed portions of the rows (SSU_g_). Each SSU_g_ consisted of four sequential sweeps performed low to high while hand-shaking the above leaves, encompassing about 1.5–2 m long of the grapevine row. The phenology of sampled grapevines was assessed through visual observation, following the BBCH [40]. Samplings on wild woody plants were carried out on randomly chosen shrubs/trees (SSU_s_) present along the perimeter of the vineyard. Each SSU_s_ consisted of 10 sweeps performed on the entire canopy or the canopy side towards the vineyard if it was not possible to perform the sweeps all around a single plant. Sweeps on wild woody plants were performed using a sweep net with a 2 m long stick. Spittlebug adult samplings were conservative; thus, collected insects were immediately released in the field after being identified. *Philaenus spumarius* adults were also sexed and, in 2017 only, their color phenotypes (color morphs) were determined according to [41,42,43].

In the 2016–2017 survey, 60 SSU_h_s (30 in inter-rows, 30 in headlands), 30 SSU_g_s, and 30 SSU_s_s were considered for each sampling date. Sampling was carried out from the first appearance of early adults in quadrat samplings (SSU_p_) until late November or when no spittlebugs were found in any SSU for two consecutive dates. In the 2018 survey, 15 SSUs in each of the three compartments were carried out twice, one in late spring (June) and the other in late summer (September). The sampling dates were chosen to roughly coincide with important phenological moments of spittlebugs in vineyards, i.e., the peak of nymphal population, the adult postemergence peak, and the late summer peak of adult population.

### 2.4. Degree-Days

In the 2016–2017 survey, air temperature was monitored to study the phenological development of *P. spumarius* nymphs [8]. A data logger (HOBO U23-002; Onset Computer, Bourne, MA, USA) was installed in a grapevine row in the center of the PSU, 50 cm above soil level and covered by a solar radiation shield. This location permitted us to obtain weather data similar to those experienced by nymphs and adults present on ground cover and grapevine canopy. Hourly air temperature data, expressed in °C, were used to elaborate degree-days (DD) according to the following formula:$$\mathrm{DD}=\overset{n}{\underset{i=1}{\sum }}\frac{max\left[0,\left(T_{i}-T_{0}\right)\right]}{24}$$
where *n* is the number of hours in one year from 1 January to 31 December, $T_{i}$ is the air temperature at hour *i*, and $T_{0}$ represents the lower developmental threshold temperature for *P. spumarius*. In this study, the lower developmental threshold temperature was set at 8 °C based on preliminary results from experiments at controlled temperatures (authors’ unpublished data) and previous literature [44,45,46].

### 2.5. Statistical Analysis

To assess the host preference of *P. spumarius*, we compared the relative frequency distribution of plant families in the sampling area (V) with the relative frequency distribution of plant families selected by nymphs of *P. spumarius* (S). V(i) is the proportion of a SSU_p_ covered by the *i-th* plant taxon out of the total of k taxa found in the SSU_p_. S(i) is the proportion of individuals found on the *i-th* plant taxon out of the total number of *P. spumarius* individuals found in the sampling area. The hypothesis is that in the absence of host preference by *P. spumarius*, S should be equal to V. For the i−th plant family, if S(i)>V(i) this plant taxon is more attractive for spittlebugs than the others, otherwise if S(i)<V(i) the plant taxon is less attractive for *P. spumarius* than the others. To measure the degree attractiveness of the i−th plant family for *P. spumarius* (KLD(i)), we applied the Kullback–Leibler divergence ($D_{KL}$) measure [47]:$$D_{KL}\left(S\|V\right)=\sum_{i=1}^{k}S\left(i\right)\ln\left(\frac{S\left(i\right)}{V\left(i\right)}\right);$$
$$KLD\left(i\right)=S\left(i\right)\ln\left(\frac{S\left(i\right)}{V\left(i\right)}\right);\;KLD\left(i\right)\in \left(-\infty ,+\infty \right).$$

The analysis of host preference on nymphal-stage sampling was conducted by pooling the data of 2016 and 2017, given that preliminary analyses performed separately on data from the two years showed no significant differences in plant association despite the mowing treatment that occurred in 2017 (see, for example, Appendix A). Adult-stage association to specific plants was not carried out, because adults both have high mobility and can be sampled on plants on which they do not feed upon. Furthermore, sampling using sweep nets does not allow sampling of specific plants, especially on herbaceous cover.

To investigate if the number of spittlebug nymphs (response variable) was different according to *Zone* (inter-rows and headlands), *cover percentage* (continuous), and *mean height* (continuous) of herbaceous cover (covariates), a negative binomial GLMM (package MASS) was performed on the 2016 data only (given the mowing disturbance during 2017 survey), with *sampling date* as the random intercept. Sex ratio through sampling season was analyzed by binomial GLMM (package lme4), with fixed variables *Season* (levels: April–June, July–September, October–December), *Vegetation compartment*, *Year*, and their interactions, and *Date* as random factor. Variations in color morph frequency throughout 2017 were analyzed by negative binomial GLM with *Sex*, *Season*, *Vegetational component*, and their interactions as fixed effects. Model assumptions were verified by plotting residuals versus fitted values, versus each covariate in the model. The packages lme4 and MASS in the software R [48,49,50] were used to fit the models.

## 3. Results

### 3.1. Species Composition

Three true spittlebug species were identified in the first survey (2016–2017 in Asti): *P. spumarius*, *N. campestris* (Fallen), and *Aphrophora alni* (Fallen). *Philaenus spumarius* was the predominant species, contributing to 80.6% of total spittlebug nymphs (*n* = 6383, 7.45 ± 0.39 nymphs/SSU_p_) and 85.9% of total spittlebug adults (*n* = 3156, 0.89 ± 0.03 adults/SSU) sampled in the 2016–2017 surveys. *Neophilaenus campestris* accounted for 18.8% of nymphs (*n* = 1490, 1.74 ± 0.17 nymphs/SSU_p_) and 10.8% of spittlebug adults (*n* = 396, 0.11 ± 0.01 adults/SSU). *Aphrophora alni* accounted for 0.6% of the total nymphs and 3.27% of total adults (*n* = 120). Another Cercopoidea species (*Cercopis vulnerata* (Rossi), (Hemiptera: Cercopidae)) was occasionally sampled as adult on the herbaceous cover (*n* = 15). The xylem-sap feeder sharpshooter *Cicadella viridis* (Hemiptera: Cicadellidae: Cicadellinae) was locally abundant on the herbaceous cover, especially in humid lower areas of the headlands (95% of total individuals, *n* = 231), while only two individuals were collected on grapevines. The planthopper *Agalmatium flavescens* (Olivier) (Hemiptera: Issidae) was quite abundant on wild woody plants (*n* = 57).

### 3.2. Nymphal Stages of Spittlebugs

#### 3.2.1. Abundance and Phenology

Nymphal stages of spittlebugs were observed on the herbaceous cover of the vineyard agroecosystem from early March through early–mid-May (Figure 2 and Appendix A). *Philaenus spumarius* was the most abundant species, reaching an average density per sampling date of ≈60 nymphs/m^2^ (IQR: 43.7–76.2) from the beginning of April to early May, roughly corresponding to the period when all the nymphs had already hatched and before the adult emergence. The registered maximum average density of *P. spumarius* was 76.5 nymphs/m^2^ on 27 April 2016, and the maximum number of individuals in a single quadrat was 158 (corresponding to a density of 632 nymphs/m^2^) on 5 May 2016. In 2017, after the mowing treatment in mid-April (12 April 2017), the densities collapsed from 70.8 to 5.6 (inter-rows) or 10 (headlands) nymphs/m^2^, remaining low until adult emergence.

*Neophilaenus campestris* nymphs were, on average, more abundant in inter-rows (30.1 nymphs/m^2^, IQR 26.6–34.7) than in headlands (8.5 nymphs/m^2^, IQR 6.7–9.2) from the beginning of April to early May 2016 (LMM: *Zone* χ^2^ = 15.99, df = 1, *p* < 0.001), possibly because of the higher abundance of Poaceae inside the vineyard (Figure 3). The maximum density of *N. campestris* was 128 individuals in a single quadrat (corresponding to a density of 512 nymphs/m^2^) on 12 April 2016. In 2017, the nymphal population in the vineyard agroecosystem was lower than in 2016, even before the mowing treatment was carried out in April. *Aphophora alni* nymphs were seldom sampled, being found mostly in localized zones of headlands, and population densities were rarely above one nymph/m^2^ (Appendix A). Nymphs of *N. campestris* and *A. alni* showed differences in their spatiotemporal distributions, with the former being more abundant in inter-rows with a dramatic decrease of population level in 2017 compared to 2016, and the latter—although with very low population levels—being more abundant in headlands and in the 2017 survey.

#### 3.2.2. Chronological and Physiological Timing of *P. spumarius* Nymphs

In 2016, sampling started in late March (23 March). During the first sampling, *P. spumarius* instars from 1st to 3rd were already present; thus, the appearance of the first nymphs was missed. In 2017, the appearance of first instar nymphs was observed from the second week of March (82 DD), the seasonal peak of nymphal populations occurred in mid-April (232 DD on 12 April), and last nymphs were observed in early May (382 DD). The time span of the nymphal stages, i.e., from the appearance of the first nymphs to the appearance of the first adults, was 54 days (254 DD) (Figure 4).

#### 3.2.3. Selection and Exploitation of Host Plants

The nymphs of the three spittlebug species differed greatly in plant association. *Philaenus spumarius* was found mainly on Asteraceae (1745 nymphs, representing 27.5% of total individuals collected), Plantaginaceae (1230 nymphs: 19.3%), and Fabaceae (1066 nymphs: 16.8%), with other botanical families less represented, e.g., Caryophyllaceae (11.8%), Poaceae (6.1%) (Table 1). Nymphal instars were unevenly distributed on different plant families. Early instars (1st to 3rd) were more frequently observed on Asteraceae (mostly on *Taraxacum officinalis* L. and *Hypochaeris* spp.), Plantaginaceae (*Veronica arvense* L. and *Plantago lanceolata* L.), and Caryophyllaceae (*Cerastium glomeratum* Thuill.). Later instars (4th and 5th) increased their exploitation of Fabaceae (mainly *Trifolium* spp. and *Medicago* spp.), Poaceae, and Polygonaceae (Figure 5 and Figure 6). The life cycles of plants influenced shifts of host selection by spittlebugs during spring. For example, Caryophyllaceae, due to their short vegetative cycle [51], hosted up to 15–18% of the total nymphs observed until mid-April in 2016 (no mowing disturbance). The steep increase of nymph percentage observed on other botanical families later in the season, corresponding to a decrease of those found on Asteraceae and Plantaginaceae, was mainly due to the increase of nymphs observed on Polygonaceae (*Rumex* spp.) and Convolvulaceae (*Convolvulus arvensis* L.). In 2017, higher plants, e.g., those belonging to Polygonaceae, were entirely cut by the mowing treatment; therefore, the basal rosettes of Asteraceae and low-height *Trifolium* hosted most of the residual spittlebug population in the vineyard (Figure 5 and Figure 6). Based on the KLD index for attractiveness, Asteraceae was the most attractive plant family (0.3) and Caryophyllaceae and Plantaginaceae were slightly attractive (0.1), whereas both Fabaceae and Poaceae exhibited low levels of attractiveness (KLD −0.11 and −0.15, respectively) (Figure 7).

Considering the botanical genus level, *P. spumarius* nymphs were observed on 62 different genera. Nearly half (46.7%) of the total individuals were collected from five genera (*Plantago*, *Cerastium*, *Trifolium*, *Taraxacum*, and *Medicago*) (Figure 6 and Appendix A). Some instars were more frequently observed on specific plant genera. For example, 49.7% of the 5th instar nymphs were collected on *Trifolium* and 61.9% on *Rumex* during the 2016 survey, while *Plantago* and *Taraxacum* hosted high percentages (20.7–28%) of 5th instar nymphs over the total nymphs found on them. Some Caryophillaeceae (*Cerastium* and *Veronica*) and *H. radicata* (Asteraceae) showed a higher colonization rate by early instars (1st to 3rd) (68–87%) compared to other genera (Figure 6).

*Neophilaenus campestris* nymphs were found almost exclusively on Poaceae (1412 nymphs; 94.8% of total nymphs observed), mostly on genera *Poa*, *Bromus*, *Avena,* and *Lolium*. It is worth noting that grasses were only seldom identified at the genus level, given the absence of morphological traits necessary for identification in early spring; thus, we do not provide precise nymph–host plant association data at the genus level. Only occasionally were *N. campestris* nymphs found on dicotyledon plants (5.2% of total nymphs), mainly on Asteraceae (1.8%) and Fabaceae (1.4%) with the genera *Picris*, *Trifolium*, and *Rumex* more represented. *Aphrophora alni* nymphs were rarely observed (*n* = 49), and mainly on Poaceae (59.2%) and Asteraceae (24.5%).

In 2016, the density of *P. spumarius* nymphs was slightly higher in SSU_p_s with high plant heights: the predicted number of nymphs/m^2^ with a grass height of 33 cm (+1 SD) was 285 ± 35.5, compared to 235 ± 27.4 nymphs with a grass height of 18 cm (i.e., the average grass height in the survey) (GLMM: average height of vegetation: χ^2^ = 6.46, df = 1, *p* = 0.011). The density of *P. spumarius* nymphs was slightly higher in headlands than in inter-rows (GLMM: zone: χ^2^ = 3.61, df = 1, *p* = 0.057). The percentage of herbaceous cover did not significantly influence the abundance of nymphs, with some quadrants with less than 25% of herbaceous cover hosting a high number of nymphs (>200/m^2^) (GLMM: percentage of herbaceous cover: χ^2^ = 1.77, df = 1, *p* = 0.183) (Appendix A). Conversely, as reported above, *N. campestris* nymphs were less abundant in headlands than in inter-rows, especially where the height of herbaceous cover was high. Indeed, the predicted density of nymphs/m^2^ when grass height increased from 18 to 33 cm lightly decreased from 148.7 ± 26.7 to 127 ± 27.6 in inter-rows, while it dropped from 78.0 ± 16.2 to 32.8 ± 8.6 in headlands (LMM: average height of vegetation × Zone: χ^2^ = 13.04, df = 1, *p* < 0.001). Furthermore, the density of *N. campestris* nymphs was slightly higher in quadrants with a higher percentage of herbaceous cover, especially in headlands, where an increase from 85% to 100% of herbaceous cover led to an increase of spittlebug density from 68.8 ± 14.2 to 113.5 ± 29.4 nymphs/m^2^ (LMM: % of herbaceous cover × zone: χ^2^ = 5.6, df = 1, *p* = 0.018) (Appendix A).

### 3.3. Adults

#### 3.3.1. Abundance and Temporal Dynamics

Teneral spittlebug adults, i.e., just emerged, were observed in sampling quadrats (SSU_p_s) from late April. Adults of all the three spittlebug species found during nymphal samplings—*P. spumarius*, *N. campestris,* and *A. alni*—were collected in the vineyard agroecosystem (PSU). The emergence of adults from nymphal stages corresponded to population peaks in all the vegetation compartments investigated. In mid- and late May, the density of *P. spumarius* adults was the highest among those observed during the year in both grapevines (up to 0.43 ± 0.07 adult/sweep) and wild woody plants (0.48 ± 0.06 adult/sweep), while on herbaceous cover the spring population peak was lower (≈0.4–0.7 adults/sweep) than the abundance registered in late summer/early autumn (up to 2 adults/sweep) (Figure 8). The pattern of *P. spumarius* density (insects/sweep) was similar on grapevines and wild woody plants, i.e., after the spring peak (0.4–0.2 insect/sweep), the abundance steadily decreased during the summer to a level of 0.1–0.2 insect/sweep, dropping close to zero in late September/early October (Figure 8). A different density pattern was observed on the herbaceous cover; indeed, after the postemergence peak (0.2 insect/sweep), the abundance dropped during summer months, newly increasing in September, reaching up to 0.8 insect/sweep, and then decreasing again in autumn due to the natural mortality of spittlebug adults. The last adults of *P. spumarius* were collected on the herbaceous cover in mid-December. In 2017, the abundance of meadow spittlebug was lower than in 2016 in all vegetational compartments. In particular, the number of adults peaking in late spring on grapevine and wild woody hosts and in September on the herbaceous cover was half that of 2016 results (Figure 8). The adult density of *P. spumarius* on grapevine peaked during the crop phenological states of late inflorescence developing and early flowering (BBCH: ≈55–65). The spittlebugs were present, although at a lower density level, through all the summer growth stages of grapevine, disappearing from the crop as soon as it reached the senescence stage (BBCH: ≥ 90).

*Neophilaenus campestris* adults were only sporadically collected on grapevines and wild woody hosts, mainly in the late spring postemergence period, and the abundance was about one-fifth or one-tenth, respectively, of that of *P. spumarius* (0.02–0.1 adult/sweep) (Figure 9). Conversely, on herbaceous cover the seasonal dynamic was similar to that observed for *P. spumarius*, with two peaks in spring and late summer/early autumn, reaching the maximum average density of 0.12 insect/sweep in late September (Figure 9).

*Aphrophora alni* adults were consistently present on wild woody vegetation from emergence in late spring until early autumn, reaching density peaks of about 0.03 individuals/sweep in June–August, while they were rarely sampled on grapevines and herbaceous cover.

#### 3.3.2. Distribution on Woody Plants

Oaks (*Quercus* spp., mainly *Q. petraea*) represented the preferred plant taxon (34% of total adult samples), with the highest density of all spittlebug species. A total 56.9% of *P. spumarius* adults sampled on wild woody plants (*n* = 586) were collected on *Quercus*, while *Prunus* spp. (*P. avium* L., *P. serotina* L., *P. padus* L.), *Cornus sanguinea* L., and *Robinia pseudoacacia* L. accounted for about 5–6% each (Table 2). A few individuals of *N. campestris* (*n* = 16) were collected on oaks and *Robinia pseudoacacia* L., while *A. alni* adults (*n* = 97) were sampled mainly on oaks, *Prunus* spp., and elm (*Ulmus minor* Mill.). Elm was particularly preferred by *A. alni*, given that 20% of total *A. alni* was sampled on this host plant, even if only 6.6% of total samplings were carried out on this tree species (Table 2).

#### 3.3.3. Sex Ratio

The sex ratio of *P. spumarius*, determined over the whole season, was female-biased in both years (female proportion 2016: 0.64; 2017: 0.58). The sex ratio differed among seasons (GLMM: *Season*: χ^2^ = 24.39, df = 2, *p* < 0.001) and year (GLMM: *Year*: χ^2^ = 6.91, df = 1, *p* = 0.008), without significant interactions among variables (Appendix A). It was closer to balance during the postemergence period (April–June), after which the male proportion declined throughout summer and autumn. The sex ratio was particularly female-biased from September on herbaceous cover and grapevines (female:male ≈ 2:1); however, no significant differences in sex ratio among vegetation compartments were observed (GLMM: *Vegetation compartment*: χ^2^ = 0.89, df = 2, *p* = 0.61; *Vegetation compartment* × *Year*: χ^2^ = 5.12, df = 2, *p* = 0.077).

#### 3.3.4. Color Morphs of *Philaenus spumarius* Adults

The most common color morph of *P. spumarius* was *typicus* (TYP: 74.7%), followed by *populi* (POP), *marginellus* (MAR), *flavicollis* (FLA), and *vittatus* (VIT). Other color morphs (*lateralis* (LAT), *leucophtalmus* (LOP), *quadrimaculatus* (Qua), *praestus* (PRA)) were rarely sampled (*n* = 27, 2.4% of total). Frequencies of color morphs differed between sexes (Color Morph × Sex: χ^2^ = 81.21, df = 8, *p* < 0.001) (Appendix A and Appendix A). Melanic morphs (MAR, FLA) were almost exclusively observed for females, while POP and VIT were more frequent in males (Appendix A). No significant differences of color morph frequencies among seasons were observed (Color Morph × Season: χ^2^ = 31.59, df = 16, *p* = 0.011). However, the color morph POP was more common during the early season (April–June: 21.5%) than in the later season (7.8–1%), a trend partially caused by the decline of the male proportion throughout the year. A lower proportion of the POP color morph later in the year was also observed in females, in favor of an increase of the MAR color morph (Appendix A). No significant differences of color morph frequencies among vegetation compartments were observed (GLMM: *Color Morph* × *Vegetation Compartment*: χ^2^ = 19.98, df = 16, *p* = 0.22).

### 3.4. The 2018 Survey

*Philaenus spumarius* was the most abundant spittlebug species in all the three monitored vineyards at both the nymphal and the adult stages (Table 3 and Table 4). *Neophilaenus campestris* was not observed in the CdA vineyard and presented a nymphal density between one-half and one-third of that registered for *P. spumarius* in the other two vineyards (AT and PD). *Aphrophora alni* nymphal stages were not observed in any vineyard, and adults were sporadically sampled on wild woody plants only. The abundance of *P. spumarius* greatly varied among the three vineyards, with PD presenting a higher abundance than the other two vineyards, especially considering the nymphal stages (Table 3). The abundances of adult spittlebugs on different vegetation compartments and periods were quite similar among vineyards and presented a pattern similar to the ones observed in Asti in the 2016–2017 surveys (Table 4). *Philaenus spumarius* was the only species present on grapevines at both time points (June and September). On herbaceous cover, adults were more abundant in September than in June, while on wild woody plants, differences in population abundances were observed between vineyards. The very high density of nymphal stages in the Paderna vineyard (13-fold higher than in Asti, Table 3) resulted in a higher adult density in all the vegetation compartments, especially on grapevines (Table 4).

## 4. Discussion

Spittlebugs are the key vectors of *Xf* in Europe, being responsible for the spreading of the bacterium in olive, almond, and Mediterranean garrigue [5,8,26,52]. *Xf* subsp. *fastidiosa* ST1 is responsible for Pierce’s disease on grapevines in North America, and it is currently present in Europe in Mallorca Island (Spain), although with low damage to local wine production [24,26]. A successful establishment of *Xf* subsp. *fastidiosa* in southern European vineyards is considered feasible based on its climatic and ecological requirements [53,54].

In this work, a three-year investigation on the abundance and phenology of both nymphal and adult stages of spittlebugs in organic vineyards in the Piedmont region of Italy was carried out to provide detailed information on spittlebugs’ timing and density (nymphal and adult stages) in different vegetation compartments throughout the year, and on host-plant association and ecological traits within vineyards located in complex landscape scenarios.

Phenological and density data collected in the two surveys showed some major outcomes: (i) *P. spumarius* was the most abundant spittlebug species at both the nymphal and adult stages, and the peak of its nymphal stages occurred from late March to late April; (ii) *P. spumarius* adults were quite abundant on grapevines in late spring (May–June); (iii) wild woody plants in the surrounding areas acted as reservoirs for spittlebug (*P. spumarius* and *A. alni*) populations during summer months; (iv) herbaceous plants in headlands presented a high density of spittlebugs (*P. spumarius* and *N. campestris*) in late summer, mainly ovipositing females; (v) grass mowing of inter-rows and partially of headlands in April drastically reduced the number of nymphs, although the subsequent adult density in the agroecosystem appeared to be less affected. In Piedmont vineyards, *P. spumarius* nymphs reached average densities of 70–75 individuals/m^2^, higher than the densities observed in Apulian, Greek, and Spanish olive groves (max 30–40 nymphs/m^2^) [6,8,36,55], and similar to those registered in one Ligurian olive grove in 2017 [8]. The nymphal abundance of *P. spumarius* observed in Californian vineyards was much lower, i.e., ≈7–11 spittlemasses/m^2^ [16].

Regarding the phenological pattern observed in vineyards, the emergence of the first nymphs occurred much earlier than has been observed in olive agroecosystems [36]. In the Asti vineyard, most of the emergence was completed at 82 DD, while completion of the emergence in olive orchards required more than 200 DD. Despite the differences between the first emergences, the pattern of emergence of other preimaginal stages was quite similar in the two agroecosystems. The variability of the thermal requirements in the DD for each stage starting from the second nymphal instar observed in vineyards is in the same range of variability as the one observed in olive orchards. These results support the hypothesis that the delayed emergence from diapause in the Mediterranean climate in the subcontinental climate, where the Asti vineyard is located, is mainly due to the diapause termination. The mechanism for diapause termination in *P. spumarius* is indeed influenced by the duration of low temperatures and photoperiod [56,57], with longer cool periods favoring faster embryonal development. However, the precise thermal requirement for the emergence of the first nymphs is still not known (however, see [44]). Results showed that the DD linear model cannot account for the differences in the DD accumulation that occurred in the two agroecosystems. Therefore, a temperature-dependent nonlinear model is probably required to describe the diapause of *P. spumarius*. This nonlinear model could account for possible adaptive responses of the species to different climatic conditions, anticipating the emergence in more continental climates than warmer ones like the one in the Mediterranean basin.

The pattern of *P. spumarius* adult density was characterized by two peaks on herbaceous cover in spring (corresponding to adult emergence) and in late summer/early autumn (corresponding to the oviposition period). Similar trends have been observed in vineyards in northeastern Italy [32] and different ecosystems in the Mediterranean area [8,58]. During the summer period, the abundance on herbaceous cover was low. However, spittlebug adults were quite abundant on the surrounding wild woody plants, similar to the trend observed in Ligurian olive groves, but different from the pattern observed in southern Italian, Spanish, and Greek olive groves, where in summer months the spittlebug population is often nearly absent in all the vegetation compartments [6,8,59]. Adult densities recorded during the 2016–17 survey were higher than those reported in most of the studies in European and Californian vineyards (e.g., [6,20]), but similar to the highest ones registered in Mediterranean olive groves (Apulia) (0.3–0.7 individuals/sweep). In other North American agroecosystems, higher densities of *P. spumarius* have been reported [39,60]. On grapevines, adults of *P. spumarius* were sampled from late May to late September, generally at low densities (0.1 individuals/sweep), although peaks over 0.4 individuals/sweep were observed in late spring of 2016. A peak in abundance of spittlebugs on grapevines was reported by [32] in the same period. Wild woody hosts in the surroundings of the vineyard hosted quite a high population of the meadow spittlebug from emergence till September. *Philaenus spumarius* detection probability may be influenced by environmental variables, e.g., forest coverage and weed height [31]. However, the variations in density within and among vegetation compartments registered throughout this study did not seem to significantly suffer from under detection of spittlebugs, since the observed population patterns differed between vegetation compartments and were consistent with those expected by the current knowledge of spittlebug ecology.

The observations made during the 2018 survey in three vineyards substantiated the general trend in phenology and host shifting observed during the first two years of survey in the Asti vineyard. The population dynamics and host shifting observed in the vineyards were quite similar to the ones recorded in olive agroecosystems, confirming the summer preference of *P. spumarius* for woody hosts (cultivated and wild) [8,61]. However, in vineyards spittlebugs seem to move mainly among vegetation compartments within the agroecosystem, whereas in dry Mediterranean olive groves they tend to leave the whole agroecosystem in summer, probably moving to oversummering host plants that are more tolerant to water stress or in more humid areas [62].

Mowing of inter-rows and headlands occurred in 2017 reduced the nymphal density 3- to 7-fold, while the adult population was approximately halved in respect to the densities observed in 2016 when no mowing was performed. Colonization of adults from unmanaged and wild surroundings, where the spittlebug population level remained high throughout our study, may have been the cause for this lower effect of mowing on the adult population compared to the effect on nymphal stages.

Host-plant association data confirmed the extreme polyphagy of *P. spumarius* nymphs and the strict association of *N. campestris* with Poaceae [38,61,63,64]. Foams of *P. spumarius* nymphs were observed on different plant families, with the most preferred being Asteraceae, Plantaginaceae, and Caryophyllaceae, and on more than 60 plant genera. Nymphs tended to shift their plant association based on their instar and time/plant phenology, moving to plants in preflowering or flowering stages, with an increase of individuals found on Fabaceae, Polygonaceae, and Poaceae after mid-April. This plant association trend has some similarities with trends observed in previous studies. Additionally, some important differences emerged. First, the patterns of vegetation communities were different. In olive groves, Asteraceae was highly predominant in Liguria, Asteraceae, and Fabaceae in Apulia and Portugal, and Asteraceae, Apiaceae (*Daucus*), and Geraniaceae (*Erodium*) in Spain [6,8,36,55]. Association with plant genera or species is even more spatially and temporally variable, given that plant communities may drastically change through time and among sites (e.g., [65]). The density of spittlebug nymphs was also influenced by the height of herbaceous cover. However, *P. spumarius* and *N. campestris* responded differently, with the former being more abundant and the latter less abundant on high grasses. *Neophilaenus campestris* was also more abundant in inter-rows, particularly where herbaceous cover was dense, probably because inter-rows were more favorable to monocots compared to headlands. Adults of *P. spumarius* sampled on wild woody hosts were particularly abundant on oaks surrounding the vineyard, similar to what was observed in Mediterranean olive groves and other ecosystems, highlighting a preference for *Quercus* spp. [6,61,62]. The great differences in spittlebug density among crops, sites, species, and vegetation emphasize the need for focused studies for each agroecosystem at risk of *Xf* introduction to provide valuable information on spittlebug distribution and to develop control strategies based on plant community management. To our knowledge, this is the first study to provide quantitative data on the plant associations of spittlebug in the vineyard agroecosystem.

To summarize, spittlebugs may be abundant in vineyards, complete their cycle within the agroecosystem, and visit grapevines, especially in late spring. Since previous contributions [19,20,26,66] stated that *P. spumarius* feeds on and transmits *Xf* to grapevines, it can be concluded that the possible introduction and establishment of a *Xf* genotype able to infect grapevines would likely be followed by the spread of the pathogen in European vineyards. Several aspects, besides vectors’ presence and abundance, need to be considered to predict possible scenarios in case of *Xf* introduction in European—and more specifically northern Italian—vineyards, namely (i) vineyard and grapevine management; (ii) landscape structure (e.g., presence of wild woody plants in the surroundings, homogeneity); (iii) *Xf* subsp. *fastidiosa* characteristics; and (iv) vector–pathogen interactions and dynamics.

The Asti vineyard represents an epitome of an organic and low-input vineyard located within a complex agricultural landscape. These conditions are expected to represent the optimal situation for high levels of resident populations of spittlebugs, compared to more managed agroecosystems and less diverse landscapes [8,60,67,68]. Thus, similar management and spatial characteristics probably correspond to the worst-case scenario for spittlebug abundance, as the 2018 survey suggests. Indeed, the highest spittlebug density was recorded in a vineyard (PD) surrounded by a complex landscape constituted by wild or low-management areas, while the lowest density was recorded in a vineyard (CdA) located in a more managed agricultural landscape and almost completely surrounded by a road. Conversely, spittlebug densities observed in other wine-producing areas (e.g., Mallorca, Apulia) were far lower than the ones registered during the surveys we carried out or in most olive groves, probably due to the generally more intense pest and weed control in vineyards than in olive and almond groves [26,67]. However, current measures against grapevine-sucking insects, e.g., insecticide application against the vector of Flavescence dorée *Scaphoideus titanus* Ball, would not be effective in controlling the spittlebugs and prevent *Xf* transmission, because adults colonize the grapevine canopy well before scheduled treatments (from the end of June onwards).

Timing of *Xf* transmission to the vine is crucial, since early infections, i.e., occurring before May–June, have a higher probability of determining systemic infections and related economic damage. Conversely, later *Xf* inoculations are associated, at least in California, with high rates of natural recovery due to winter chilling [69,70]. The observed abundance peak of *P. spumarius* on grapevines in May–June would allow the establishment of chronic *Xf* infections if the pathogen was introduced into the agroecosystem. This paradigm is, however, under scrutiny, and milder winters caused by climate change could reduce the induction of plant recovery [20,70].

Furthermore, *P. spumarius* could be able to transmit *Xf* from vine to vine (secondary spread), given its relative preference for grapevines, its long adult life, and the *Xf* transmission biology (e.g., persistent without a latent period) [20,66,71]. However, further studies on movements between host plants and feeding behavior are needed to assess the risk of secondary spread, since the preferred feeding sites on grapevines of the spittlebug could be located on the distal portions of stem, where *Xf* populations are usually low until midsummer, thus impairing efficient vine-to-vine transmission in the early season, the optimal period for establishing systemic infection [70]. Climate change could also play a two-faced role in shaping future *Xf*–grapevine scenarios in Europe as, on one hand, it could make several wine-producing regions more suitable to *Xf* establishment [53], while on the other, it could determine a northward shift of *P. spumarius* distribution, decreasing the suitability of southern European areas for this insect [72].

Further studies on spittlebugs in European vineyard agroecosystems are needed to clarify the aforementioned aspects of vector–pathogen–host dynamics from the perspective of reliable risk assessment of *Xf* establishment and spread, and to develop effective control measures against insect vectors to limit the pathogen expansion and damage to this crop.

## Figures and Tables

**Figure 1 insects-12-01012-f001:**
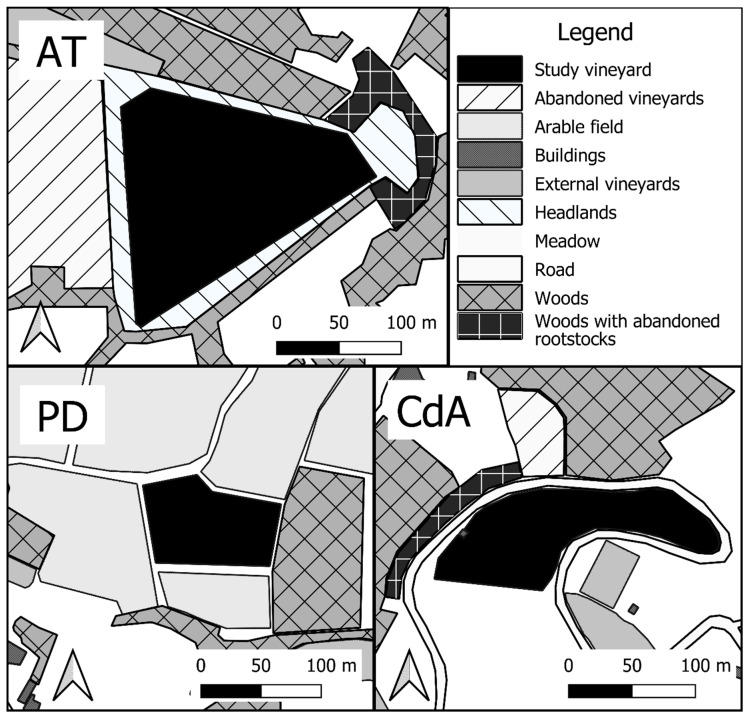
Vegetal composition of the surroundings of the three surveyed vineyards: AT: Asti; CdA: Cisterna d’Asti; PD: Paderna.

**Figure 2 insects-12-01012-f002:**
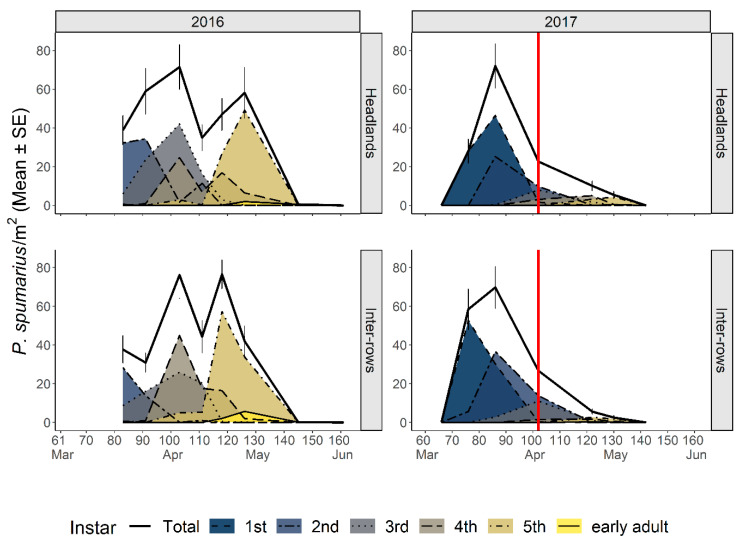
Life-stage structure of the preimaginal populations of *Philaenus spumarius* in vineyard inter-rows and headlands. The area curves of each nymphal instar, the total nymphs, and the newly emerged adults are represented as mean densities by sampling date. Red vertical lines represent the date of mowing treatment. Data refer to the 2016–2017 survey in Asti.

**Figure 3 insects-12-01012-f003:**
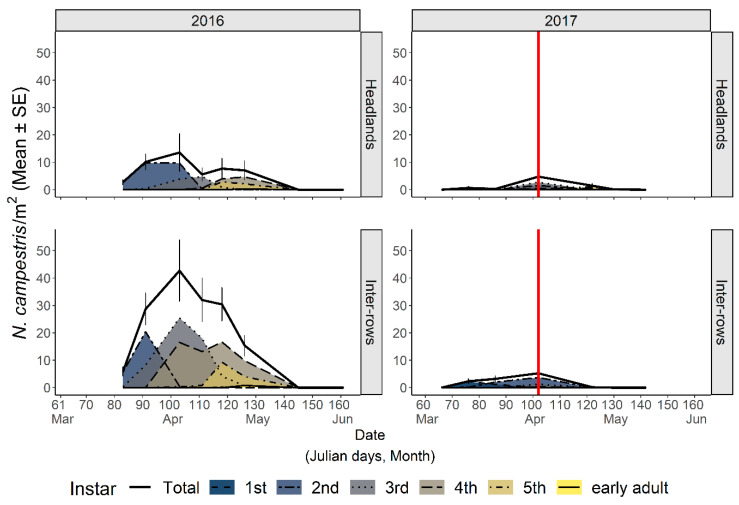
Life-stage structure of the preimaginal populations of *Neophilaenus campestris* in vineyard inter-rows and headlands. The curves area of each nymphal instar, the total nymphs, and the newly emerged adults are represented as mean densities by sampling date. Red vertical lines represent the date of mowing treatment. Data refer to the 2016–2017 survey in Asti.

**Figure 4 insects-12-01012-f004:**
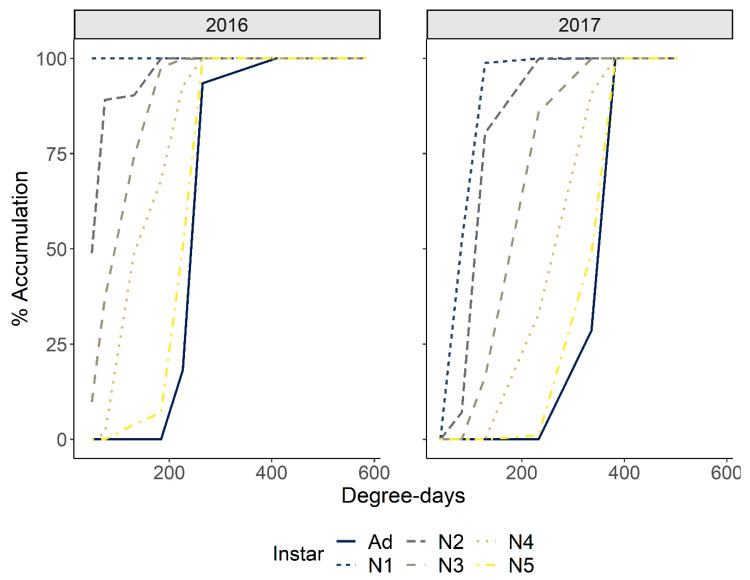
The observed cumulative population of the nymphal instars and newly emerged adults of *Philaenus spumarius* according to accumulated degree-days in the Asti (northwestern Italy) vineyard in 2016–2017.

**Figure 5 insects-12-01012-f005:**
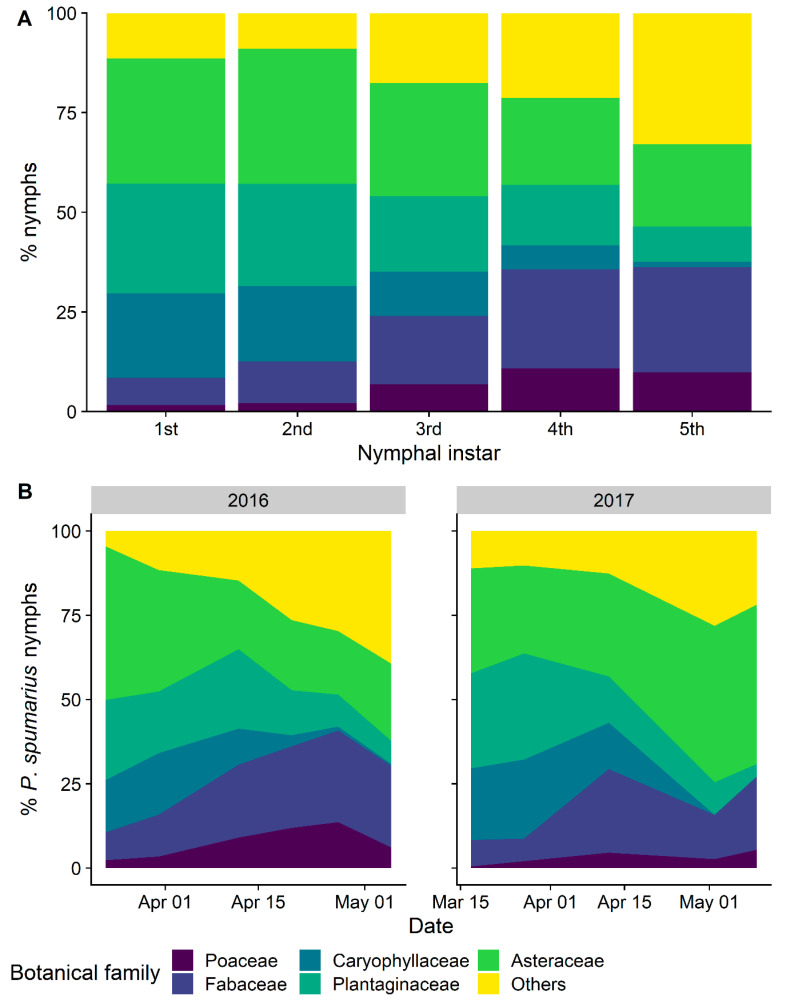
Relative percentages of nymphs of *Philaenus spumarius* on the most selected plant families in vineyard in the 2016–2017 samplings. (**A**) per instar; (**B**) throughout the sampling period (March–May).

**Figure 6 insects-12-01012-f006:**
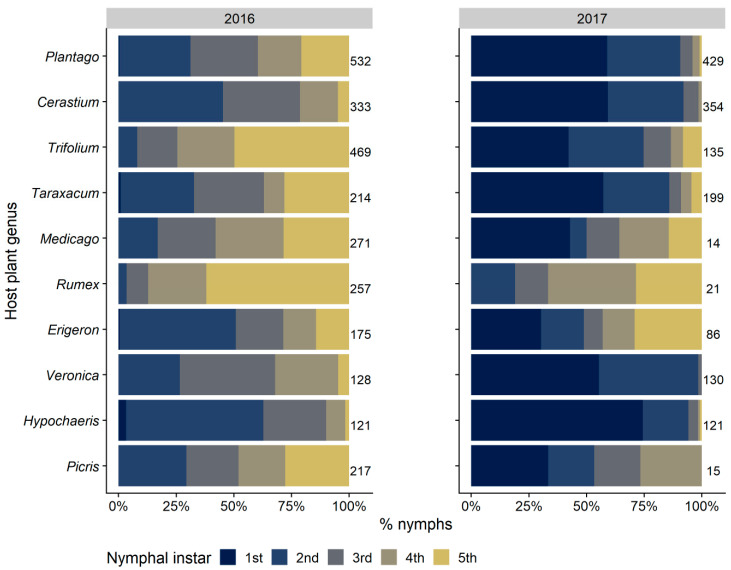
Relative distribution (percentages) of the nymphal instars of *Philaenus spumarius* on each host plant genus. The 10 host plant genera in vineyard hosting the highest number of spittlebug nymphs are shown. Numbers beside bars represent the total number of nymphs sampled on each genus by year of survey: 2016 (**left**) and 2017 (**right**).

**Figure 7 insects-12-01012-f007:**
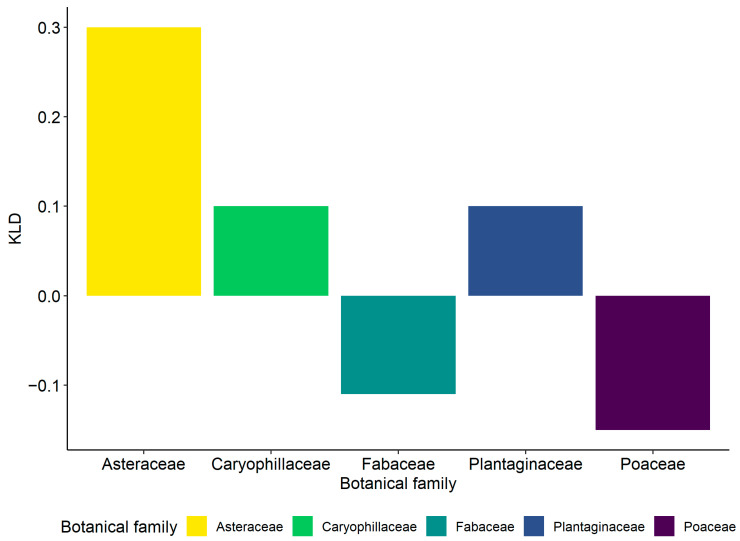
Kullback–Leibler (KLD) index estimating the level of attractiveness of the most common plant families in the 2016–17 survey for the nymphs of *Philaenus spumarius*.

**Figure 8 insects-12-01012-f008:**
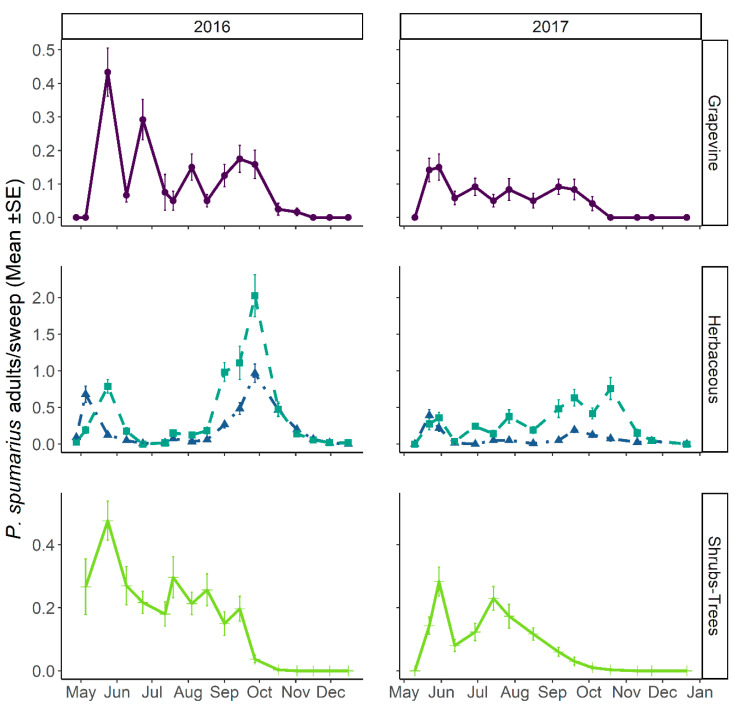
The abundance of *Philaenus spumarius* adults on grapevine (**top**), herbaceous cover (**middle**), and shrub–tree (**bottom**) components in the Asti vineyard throughout 2016 (**left**) and 2017 (**right**) sampling periods. In herbaceous vegetation, dashed lines represent headlands and dash-dotted lines represent inter-rows.

**Figure 9 insects-12-01012-f009:**
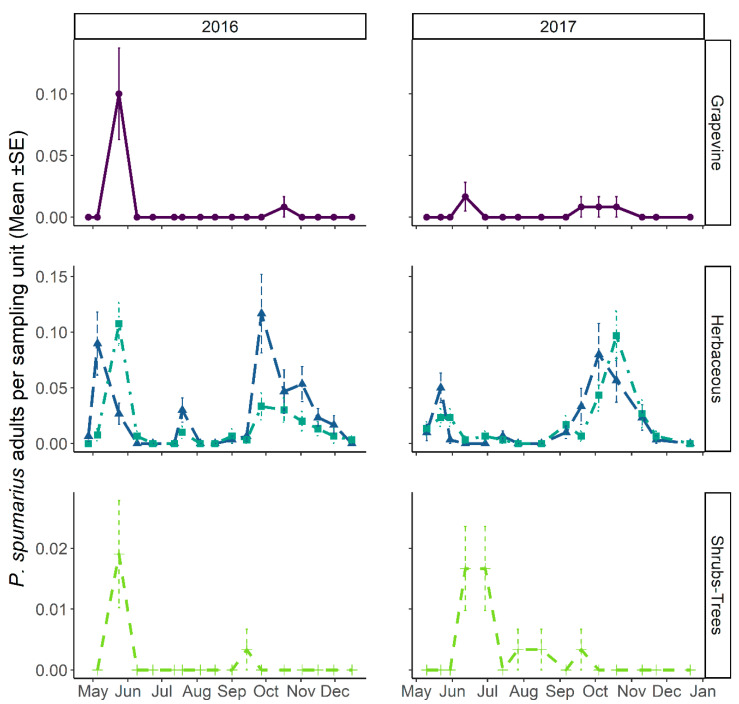
The abundance of *Neophilaenus campestris* adults on grapevine (**top**), herbaceous cover (**middle**), and shrub–tree (**bottom**) components in the Asti vineyard throughout 2016 (**left**) and 2017 (**right**) sampling periods. In herbaceous vegetation, dashed lines represent headlands and dash-dotted lines represent inter-rows.

**Table 1 insects-12-01012-t001:** Absolute (number) and relative (percentage) distributions of spittlebug nymphs according to plant families on which they were found in an organic vineyard in Asti (northwestern Italy) during the 2016–2017 surveys. Plant families are arranged in descending order of *Philaenus spumarius* nymphs.

	*Philaenus spumarius*		*Neophilaenus campestris*		*Aphrophora alni*	
Host Plant Family	No. Nymphs	%	No. Nymphs	%	No. Nymphs	%
Asteraceae	1745	27.45	27	1.81	12	25.53
Plantaginaceae	1230	19.35	9	0.60	3	6.38
Fabaceae	1066	16.77	21	1.41	1	2.13
Caryophyllaceae	753	11.85	—	—	—	—
Poaceae	386	6.07	1412	94.77	29	61.70
Polygonaceae	291	4.58	8	0.54	—	—
Rosaceae	59	0.93	—	—	1	2.13
Rubiaceae	37	0.58	—	—	1	2.13
Others	770	12.11	13	0.87	—	—
Nymphs not on plants	20	0.31	—	—	—	—

**Table 2 insects-12-01012-t002:** The abundance of spittlebug adults (total and mean individuals/SSU) on genera of wild woody plants in the surroundings of the Asti vineyard in 2016–17 surveys.

Host Plants			*Philaenus spumarius*			*Neophilaenus campestris*			*Aphrophora alni*		
Genus	No. Samples	%	*n*	Mean/SSU	%	*n*	Mean/SSU	%	*n*	Mean/SSU	%
*Quercus*	283	33.85	586	2.07	56.89	5	0.02	27.78	36	0.13	37.11
*Prunus*	58	6.94	63	1.09	6.12	1	0.02	5.56	13	0.22	13.40
*Cornus*	50	5.98	59	1.18	5.73	2	0.04	11.11	6	0.12	6.19
*Robinia*	63	7.54	55	0.87	5.34	4	0.06	22.22	3	0.05	3.09
*Ulmus*	55	6.58	46	0.84	4.47	1	0.02	5.56	19	0.35	19.59
*Crataegus*	46	5.50	41	0.89	3.98	2	0.04	11.11	7	0.15	7.22
*Pyrus*	37	4.43	37	1.00	3.59	1	0.03	5.56	1	0.03	1.03
*Morus*	18	2.15	17	0.94	1.65	0	0.00	0.00	8	0.44	8.25
*Carpinus*	10	1.20	5	0.50	0.49	0	0.00	0.00	2	0.20	2.06
*Arbusto*	2	0.24	2	1.00	0.19	0	0.00	0.00	1	0.50	1.03
*Alnus*	1	0.12	0	0.00	0.00	0	0.00	0.00	1	1.00	1.03

**Table 3 insects-12-01012-t003:** The abundance of spittlebug nymphs (total and mean individuals/m^2^) on herbaceous cover of inter-rows in three organic Piedmont vineyards (AT = Asti, CdA = Cisterna d’Asti, PD = Paderna) in 2018.

		*Philaenus spumarius*			*Neophilaenus campestris*		
Site	No. SSU_p_	Mean	SEM	*n*	Mean	SEM	*n*
AT	15	9.07	2.36	34	4.8	2.04	18
CdA	15	5.33	2.89	20	0	0	0
PD	15	138	26	516	46.1	23	173

**Table 4 insects-12-01012-t004:** Abundance of spittlebug adults (mean individuals/sweep) on different vegetation compartments in June and September in three organic Piedmont vineyards (AT = Asti, CdA = Cisterna d’Asti, PD = Paderna) in 2018.

				*Philaenus spumarius*		*Neophilaenus campestris*		*Aphrophora alni*	
Site	Month	Vegetation	*n*	Mean	SE	Mean	SE	Mean	SE
AT									
	June								
		Shrubs–Trees	15	0.06	0.02	―	―	―	―
		Inter-rows	16	0	0	―	―	―	―
		Grapevine	15	0.08	0.04	―	―	―	―
	September								
		Shrubs–Trees	15	0.11	0.04	0.02	0.01	0.01	0.009
		Inter-rows	15	0.32	0.09	0.07	0.03	―	―
		Grapevine	15	0.08	0.03	―	―	―	―
CdA									
	June								
		Shrubs–Trees	15	0.01	0.01	―	―	0.007	0.007
		Inter-rows	15	0.02	0.02	―	―	―	―
		Grapevine	15	0.05	0.03	―	―	―	―
	September								
		Shrubs–Trees	15	0	0	―	―	―	―
		Inter-rows	15	0.1	0.04	―	―	―	―
		Grapevine	15	0.02	0.02	―	―	―	―
PD									
	June								
		Shrubs–Trees	15	0.09	0.03	0.007	0.007	0.007	0.007
		Inter-rows	15	0.22	0.07	0.08	0.03	―	―
		Grapevine	15	0.3	0.06	0.017	0.017	―	―
	September								
		Shrubs–Trees	15	0.15	0.05	―	―	0.02	0.01
		Inter-rows	15	0.75	0.13	0.1	0.04	―	―
		Grapevine	15	0.42	0.07	―	―	―	―

## Data Availability

The raw data are available from the corresponding author upon request.

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
