# Peer review of "Phenology, Seasonal Abundance, and Host-Plant Association of Spittlebugs (Hemiptera: Aphrophoridae) in Vineyards of Northwestern Italy"

_insects, 2021, doi:10.3390/insects12111012_

Round 1

Reviewer 1 Report

An interesting article that provides a solid foundation for studying and managing an emerging pest, spittlebugs, in Italian Vineyards. Well written with sound scientific content. My minor suggested corrections, mostly grammatical, and included in the attached pdf as highlighted sections of text with comment boxes detailing the suggested changes.

Author Response

We appreciate the general appraisal of our work and the Reviewer’s suggestions. We changed the MS according to the suggested corrections (see revised version).

Reviewer 2 Report

Title: Phenology, seasonal abundance and host-plant association of spittlebugs (Hemiptera: Aphrophoridae) in vineyards of North Western Italy.

Overview and general recommendation:

The manuscript presents a survey carried out in three vineyards in Northwestern Italy to monitor diversity, phenology and host-plant association of nymph and adult spittlebugs. Integrating several sampling methods, the authors give a detailed description of Philaenus spumarius, Neophilaenus campestris and Aphrophora alni distribution, life stage structure and prevalent host plant families. The paper adds further knowledge on spittlebugs presence in vineyards, showing that the studied species can complete their lifecycle in the surrounding agroecosystem, providing new evidence on the potential spread of Xylella fastidiosa in grapevine.

The manuscript is in general well organized, however I believe there are a few key methodological points that must be revised, in order to give a stronger support to the conclusions. A detailed list of the can be found below.

While the title clearly describes the performed study, simple summary, abstract and introduction start with an extensive focus on Xylella fastidiosa (Xf). Even if the link to the potential vectors is made clear, I believe you should first concentrate on your main aims and results and only afterwards discuss the relevance of your results for the spread of Xf.

Line 36: “Field surveys of nymph and adult spittlebugs were performed in three vineyards of Northwestern Italy during three years.” – This statement should be revised since you did not monitor 3 vineyards for 3 years, according to the materials and methods section. In addition, I do not think that 3 sites are enough to call it a multi-site survey, term that I would remove from the manuscript.

Line 82: have found --> resulted in

Line 83: has been found --> was found (please check verb tenses in the rest of the manuscript).

Line 131: PSU, define acronym here (not in line 148).

Lines 133-135: for how long did you inspect vegetation and soil?

Line 148: is --> was

Lines 201-210: your analysis should take into account spatial autocorrelation between host plant species or prove that your data does not suffer from spatial autocorrelation.

Lines 217-226: considering the hierarchical structure of your dataset, the pseudoreplication you should manage is related to the repeated measures in each sampling unit, thus the random factor is the sampling unit, not the year. You should add year as a fixed factor. All models should be re-run with this correction and assess model assumptions again. Consider following Zuur, Alain F., and Elena N. Ieno. "A protocol for conducting and presenting results of regression‐type analyses." Methods in Ecology and Evolution 7.6 (2016): 636-645.

Figure 2 and 3: Use the same x-axis interval, to allow an effective comparison of the phenology between the two years.

Line 258: “Neophilaenus campestris nymphs were in average more abundant in inter-rows”, I suggest to test the difference rather than just discuss it, at this point and in the rest of the results.

Line 300: “In 2017, higher plants, e.g. those belonging to Polygonaceae, were entirely cut by the mowing treatment, therefore the basal rosettes of Asteraceae and low height Trifolium hosted most of the residual spittlebugs population in the vineyard (Figure 5 and 6).” Considering this statement and what can be observed from Fig. 2 and Fig. 3, you should not analyse the two years together since they represent two different conditions, not replicates of the same situation. I am afraid that the analyses that follow (Tab. 1, Fig.5-7) are not reliable since you are joining two datasets that should be analysed separately (2016: high plant host abundance, high nymph abundance, 2017: the opposite).

Fig. 6: use row data, otherwise, it is impossible to compare the relevance of the plant host species.

Lines 340-352: consider re-doing the analysis combing plant height and plant family.

Author Response

Overview and general recommendation:

The manuscript presents a survey carried out in three vineyards in Northwestern Italy to monitor diversity, phenology and host-plant association of nymph and adult spittlebugs. Integrating several sampling methods, the authors give a detailed description of Philaenus spumarius, Neophilaenus campestris and Aphrophora alni distribution, life stage structure and prevalent host plant families. The paper adds further knowledge on spittlebugs presence in vineyards, showing that the studied species can complete their lifecycle in the surrounding agroecosystem, providing new evidence on the potential spread of Xylella fastidiosa in grapevine.

The manuscript is in general well organized, however I believe there are a few key methodological points that must be revised, in order to give a stronger support to the conclusions. A detailed list of the can be found below.

We appreciate the general appraisal of our work and the Reviewer’s suggestions. We revised the manuscript according to the suggestions whenever we agreed with them. We provided a more more thoroughly description of some of the methodological points raised by the reviewer.

While the title clearly describes the performed study, simple summary, abstract and introduction start with an extensive focus on Xylella fastidiosa (Xf). Even if the link to the potential vectors is made clear, I believe you should first concentrate on your main aims and results and only afterwards discuss the relevance of your results for the spread of Xf.

We understand this concern in relation to the simple summary, that we have slightly refocused. As for the abstract and introduction, we would prefer to maintain the present structure, since in our opinion the initial focus on Xf is needed to make clear to the reader the rationale behind our study, i.e. why is it now important to study spittlebugs in Italian vineyards. In our opinion, to concentrate on our aims means focusing on Xf risk in vineyards, therefore the bacterium and its introduction in Europe has to be presented early in the paper. In such a way the introduction has a funnel structure and introduce some of the themes that will be discussed in the Discussion section, since we focused on the consequences of spittlebug ecology and phenology on Xf transmission.

Line 36: “Field surveys of nymph and adult spittlebugs were performed in three vineyards of Northwestern Italy during three years.” – This statement should be revised since you did not monitor 3 vineyards for 3 years, according to the materials and methods section. In addition, I do not think that 3 sites are enough to call it a multi-site survey, term that I would remove from the manuscript.

The reviewer is right. We tried to compress the whole study in a simple sentence but clearly the resulting statement does not describe correctly what was carried out and can be misleading. We rephrased the sentence to make clearer the description (see l. 137-140 of tracked changes MS) and removed all the multi-site survey references.

Line 82: have found --> resulted in Done

Line 83: has been found --> was found (please check verb tenses in the rest of the manuscript). Done

Line 131: PSU, define acronym here (not in line 148). Done

Lines 133-135: for how long did you inspect vegetation and soil? The aim was to count all the spittlebugs present within the SSU (obviously given unavoidable operator error). Therefore, we spent more time in the most complex SSU (higher height of herbaceous cover) compared to the ones with lower height and less herbaceous cover. However, we spent at least five minutes looking for spittlebug nymphs in each SSU. This information was added in the MS (l. 159 of tracked changes MS).

Line 148: is --> was Done

Lines 201-210: your analysis should take into account spatial autocorrelation between host plant species or prove that your data does not suffer from spatial autocorrelation.

Preliminary analysis of data excluded important autocorrelations among abundance of botanical families. For your convenience, we attach a simple correlation pairs plot (Pairs_cor_PlantFamilies_Asti.pdf) between some of the most important families as an example. This was quite expected, since at high taxonomical level (e.g. family) species with very different ecological needs may be pooled together. Moreover, the high number of replicas ensure the robustness of the KLD estimate also in case of slight autocorrelations.

Lines 217-226: considering the hierarchical structure of your dataset, the pseudoreplication you should manage is related to the repeated measures in each sampling unit, thus the random factor is the sampling unit, not the year. You should add year as a fixed factor. All models should be re-run with this correction and assess model assumptions again. Consider following Zuur, Alain F., and Elena N. Ieno. "A protocol for conducting and presenting results of regression‐type analyses." Methods in Ecology and Evolution 7.6 (2016): 636-645.

The paper of Zuur and Ieno (2016) is indeed a landmark of presentation style of statistical analysis, and we try to follow its suggestions when possible. However, there is an important aspect of the presented surveys that maybe was not sufficiently stressed in the MS: the sampling units were randomly positioned during each survey date, i.e. replica 1 in date 1 was not the same place of replica 1 in date 2.  They were thus true replicas, and were not included as random factor in the models for this reason. We added “at each sampling date” at line 157 (tracked changes MS) to make clearer this aspect to the reader.

Year could be considered as fixed effect as well, however we forgot to mention that models applied to nymphal data (abundance nymphs vs % cover and plant height) were applied to 2016 subset only, just because 2017 was unreliable due to the mowing treatment. Colour morphs data of adults were registered in 2017 only and therefore the model did not include the variable Year. However, we included Year as fixed effect for sex ratio of P. spumarius adults, and its effect was significant (l. 513-514, tracked changes MS).

Figure 2 and 3: Use the same x-axis interval, to allow an effective comparison of the phenology between the two years.

Done, thanks for pointing that out.

Line 258: “Neophilaenus campestris nymphs were in average more abundant in inter-rows”, I suggest to test the difference rather than just discuss it, at this point and in the rest of the results.

The reviewer is right, in the submitted MS we provided the results of statistical model (LMM) of the effect of Zone on the abundance of N. campestris nymphs later (l. 348 first submission MS). We now moved this result at line 354 (tracked changes MS).

Line 300: “In 2017, higher plants, e.g. those belonging to Polygonaceae, were entirely cut by the mowing treatment, therefore the basal rosettes of Asteraceae and low height Trifolium hosted most of the residual spittlebugs population in the vineyard (Figure 5 and 6).” Considering this statement and what can be observed from Fig. 2 and Fig. 3, you should not analyse the two years together since they represent two different conditions, not replicates of the same situation. I am afraid that the analyses that follow (Tab. 1, Fig.5-7) are not reliable since you are joining two datasets that should be analysed separately (2016: high plant host abundance, high nymph abundance, 2017: the opposite).

We understand the issue pointed out by the reviewer. Indeed, we decided to pool the data from the two years since preliminary analysis did not highlight any major difference in KLD estimates of plant preference between the two years of survey (see plots of KLD for separate years: Asti_KLD_separate_Years.pptx). There are probably two main reasons for this lack of differences: i) P. spumarius nymphs did not change dramatically their host plant preference following the mowing; ii) the small number of nymphs counted after the mowing treatments in 2017 were not sufficient to significantly affect the KLD estimates. We would hence prefer to maintain in the results the pooled KLD analysis that provide a comprehensive view of plant preference of P. spumarius nymphs in vineyard. We modified the M&M section, describing that we performed KLD analysis on separate years, but then we decided to pool together the two datasets (l. 270-272 of tracked changes MS).

Fig. 6: use row data, otherwise, it is impossible to compare the relevance of the plant host species.

We are sorry but we did not fully understand this comment. Is maybe the reviewer suggesting to use raw data in plot and not relative? We preferred to use relative data to allow an easier comparison of the distribution of nymphal instars within each host plant genus, but the raw number of nymphs observed on each plant genus is reported on the side of each bar, allowing the reader to directly compare the relevance of each plant. We would prefer to not change the figure.

Lines 340-352: consider re-doing the analysis combing plant height and plant family.

Thanks to pointing that out. Indeed, we did not describe thoroughly the model used. We included also plant height and Zone (inter-rows and headlands) as independent variables, but they did not contribute significantly in the model and were dropped with a step-wise selection method. We mentioned it in the revised MS (lines 277-279). The variable plant family was not included because the model was applied on vegetation data SSU-wise (whole vegetation present inside the quadrant), and not spittle-wise (single host plant).

Reviewer 3 Report

These are my main comments on the manuscript (Insects-1329099) entitled “Phenology, seasonal abundance and host-plant association of spittlebugs (Hemiptera: Aphrophoridae) in vineyards of North Western Italy” It is an interesting study examining the host-pant association of spittlebugs Xylostella vectors. Such studies provide useful results to develop appropriate pest management strategies. Generally, it a well-written study without any serious flaws or mistakes in methodology and presentation. However, details of statistical analyses are needed. I have suggested some changes in the manuscript. My proposal is to accept it for publication in "Insects" after minor revision.
A few points:
L.1: Research Article
Ls.18-19: Revise this sentence to eliminate wordiness
L.35: Place “,” after phenology
Ls.27-38: Sentence is repetitive (see lines 34-35)
Ls.48-49: Keywords should be in alphabetic order. Also, keywords serve to widen the opportunity to be retrieved from a database. To put words that already are into title and abstracts makes KW not useful. Please choose terms that are neither in the title nor in abstract.
Ls.63-66: Provide references for this sentence
Ls.68-69: Delete “among others,”
Ls.74 and 237: Delete “Cicadellinae”
Ls.77-79: Rephrase this sentence
L.93: Citation 26 not found
L.100: Change “perfomed” by “performed”
L.104: Sentence starting “Two-year survey…”
L.111: Delete “The”
L.117: Delete “possibly”
Ls.130, 165: Change “x” by “×”. Check in all manuscript.
Ls.130-131: Define SSUp and PSU
Ls.135-137: Rephrase
Ls.137 and 143: Delete “directly”
Ls.163 and 166: Explain SSUh and SSUg
L.177: Reference “Bondino et al., 2019” should be numbered according to the journal style
L.190: Delete “(author’s unpublished data”
L.191: Delete “(“
L.192: Explain SSUs
L.258: Delete “in average”
L.555: Delete “In summarize”

Author Response

These are my main comments on the manuscript (Insects-1329099) entitled “Phenology, seasonal abundance and host-plant association of spittlebugs (Hemiptera: Aphrophoridae) in vineyards of North Western Italy” It is an interesting study examining the host-pant association of spittlebugs Xylostella vectors. Such studies provide useful results to develop appropriate pest management strategies. Generally, it a well-written study without any serious flaws or mistakes in methodology and presentation. However, details of statistical analyses are needed. I have suggested some changes in the manuscript. My proposal is to accept it for publication in "Insects" after minor revision.

We are pleased that the Reviewer appreciates the importance of our study. We changed the MS according to the suggestions of the reviewer.

We thanks the reviewer for te
A few points:
L.1: Research Article Done
Ls.18-19: Revise this sentence to eliminate wordiness Done
L.35: Place “,” after phenology Done
Ls.27-38: Sentence is repetitive (see lines 34-35) The second sentence (Ls. 37-38) was deleted. Thanks for pointing that out.
Ls.48-49: Keywords should be in alphabetic order. Also, keywords serve to widen the opportunity to be retrieved from a database. To put words that already are into title and abstracts makes KW not useful. Please choose terms that are neither in the title nor in abstract. Usually it is suggested by Journals to avoid to put keywords already present in the title. However it is quite fine to place keywords already present in the abstract, since putting other words would mean (at least in our case) to use words that could be misleading being them too far away from the focus of the article.
Ls.63-66: Provide references for this sentence Done
Ls.68-69: Delete “among others,” Done
Ls.74 and 237: Delete “Cicadellinae” We prefer to maintain the nomenclature including the subfamily, since it is common use and the common name sharpshooter refer exclusively to that taxon.
Ls.77-79: Rephrase this sentence Done
L.93: Citation 26 not found Added
L.100: Change “perfomed” by “performed” Done
L.104: Sentence starting “Two-year survey…” Done
L.111: Delete “The” Referring to the specific site already mentioned earlier, the article is needed, or at least strongly suggested.
L.117: Delete “possibly” there are no data whatsoever on effect of pyrethrum on spittlebugs, although probably it may interfere, we can not be sure. We would prefer to maintain “possibly”.
Ls.130, 165: Change “x” by “×”. Check in all manuscript. Done
Ls.130-131: Define SSUp and PSU Done
Ls.135-137: Rephrase The sentence was deleted, given that it referred to data not included in the final version of the MS.
Ls.137 and 143: Delete “directly” Done
Ls.163 and 166: Explain SSUh and SSUg These sampling units are described in the text as they are presented (l. 195-205 of tracked changes MS).
L.177: Reference “Bondino et al., 2019” should be numbered according to the journal style Done
L.190: Delete “(author’s unpublished data” The journal Insects allow to refer to unpublished data
L.191: Delete “(“  Done
L.192: Explain SSUs The meaning of abbreviation of sampling units is described earlier (l. 166 tracked change MS).
L.258: Delete “in average” We would prefer to maintain the reference to the average, since single SSU in headlands may host higher spittlebug population than SSU in inter-rows, but the overall average was different. Following recommendation of another reviewer, we changed “in average” in “on average”
L.555: Delete “In summarize” We would prefer to maintain “to summarize”, given that it is a common signal to the reader to introduce the conclusions and the end of discussion.

Round 2

Reviewer 2 Report

The authors have revised the manuscript and I appreciate all the effort they made to respond to the expressed concerns. There is still a few small changes to do before it is ready for publication.

Modify the x-axis for Fig. 3 as it has changed for Fig. 2.

Line 54: color morphs, grapevine; --> color morphs; grapevine;

Line 252: random variable --> random factor

Moreover, include in the Supplementary Content the plots in your cover letter to allow readers understanding your data, as well as the full list of performed models (both GLMs and GLMMs) with regression parameters, since what you included in the manuscript is just the p-value of a few variables.

Author Response

Dear Reviewer,

please find point-by-point responses to your requested changes:

Modify the x-axis for Fig. 3 as it has changed for Fig. 2. Done

Line 54: color morphs, grapevine; --> color morphs; grapevine; Done

Line 252: random variable --> random factor Done

Moreover, include in the Supplementary Content the plots in your cover letter to allow readers understanding your data, as well as the full list of performed models (both GLMs and GLMMs) with regression parameters, since what you included in the manuscript is just the p-value of a few variables.

Supplementary materials have been revised and output of all the models mentioned in the MS added.